# Enrichment of Activated Fibroblasts as a Potential Biomarker for a Non-Durable Response to Anti-Tumor Necrosis Factor Therapy in Patients with Crohn’s Disease

**DOI:** 10.3390/ijms241914799

**Published:** 2023-09-30

**Authors:** Soo-Kyung Park, Gi-Young Lee, Sangsoo Kim, Chil-Woo Lee, Chang-Hwan Choi, Sang-Bum Kang, Tae-Oh Kim, Jaeyoung Chun, Jae-Myung Cha, Jong-Pil Im, Kwang-Sung Ahn, Seon-Young Kim, Min-Suk Kim, Chang-Kyun Lee, Dong-Il Park

**Affiliations:** 1Division of Gastroenterology, Department of Internal Medicine and Inflammatory Bowel Disease Center, Kangbuk Samsung Hospital, School of Medicine, Sungkyunkwan University, Seoul 03181, Republic of Korea; skparkmd@gmail.com; 2Medical Research Institute, Kangbuk Samsung Hospital, School of Medicine, Sungkyunkwan University, Seoul 03181, Republic of Korea; chilwoo.lee@gmail.com; 3Department of Bioinformatics, Soongsil University, Seoul 06978, Republic of Korea; gy48085@gmail.com (G.-Y.L.); sskib@ssu.ac.kr (S.K.); 4Department of Internal Medicine, College of Medicine, Chung-Ang University, Seoul 06974, Republic of Korea; gicch@cau.ac.kr; 5Department of Internal Medicine, Daejeon St. Mary’s Hospital, Daejeon 34943, Republic of Korea; sangucsd@gmail.com; 6Department of Internal Medicine, Haeundae Paik Hospital, College of Medicine, Inje University, Busan 47392, Republic of Korea; kto0440@paik.ac.kr; 7Department of Internal Medicine, Gangnam Severance Hospital, College of Medicine, Yonsei University, Seoul 03722, Republic of Korea; j40479l@gmail.com; 8Department of Internal Medicine, Kyung Hee University Hospital at Gang Dong, College of Medicine, Kyung Hee University, Seoul 02447, Republic of Korea; clicknox@hanmail.net; 9Department of Internal Medicine, Liver Research Institute, College of Medicine, Seoul National University, Seoul 08826, Republic of Korea; jpim0911@snu.ac.kr; 10Functional Genome Institute, PDXen Biosystems, Inc., Daejeon 34027, Republic of Korea; kwangsung.ahn@gmail.com; 11Personalized Medicine Research Center, Korea Research Institute of Bioscience and Biotechnology (KRIBB), Daejeon 34141, Republic of Korea; kimsyl@kribb.re.kr; 12Department of Human Intelligence and Robot Engineering, Sangmyung University, Cheonan 31066, Republic of Korea; minsuk.kim@smu.ac.kr; 13Department of Gastroenterology, Center for Crohn’s and Colitis, Kyung Hee University Hospital, College of Medicine, Kyung Hee University, Seoul 02447, Republic of Korea; gidrlee@gmail.com

**Keywords:** anti-tumor necrosis factor therapy, Crohn’s disease, activated fibroblasts, RNA sequencing

## Abstract

We investigated whether the response to anti-tumor necrosis factor (anti-TNF) treatment varied according to inflammatory tissue characteristics in Crohn’s disease (CD). Bulk RNA sequencing (RNA-seq) data were obtained from inflamed and non-inflamed tissues from 170 patients with CD. The samples were clustered based on gene expression profiles using principal coordinate analysis (PCA). Cellular heterogeneity was inferred using CiberSortx, with bulk RNA-seq data. The PCA results displayed two clusters of CD-inflamed samples: one close to (Inflamed_1) and the other far away (Inflamed_2) from the non-inflamed samples. Inflamed_1 was rich in anti-TNF durable responders (DRs), and Inflamed_2 was enriched in non-durable responders (NDRs). The CiberSortx results showed that the cell fraction of activated fibroblasts was six times higher in Inflamed_2 than in Inflamed_1. Validation with public gene expression datasets (GSE16879) revealed that the activated fibroblasts were enriched in NDRs over Next, we used DRs by 1.9 times pre-treatment and 7.5 times after treatment. Fibroblast activation protein (FAP) was overexpressed in the Inflamed_2 and was also overexpressed in the NDRs in both the RISK and GSE16879 datasets. The activation of fibroblasts may play a role in resistance to anti-TNF therapy. Characterizing fibroblasts in inflamed tissues at diagnosis may help to identify patients who are likely to respond to anti-TNF therapy.

## 1. Introduction

Inflammatory bowel disease (IBD), which primarily consists of ulcerative colitis (UC) and Crohn’s disease (CD), is a chronic relapsing intestinal disease that is multifactorial and polygenic, resulting from a dysfunctional epithelial barrier that finally leads to an aberrant immune response to intestinal bacteria [1]. Thus far, therapies for IBD have mainly focused on targeting immune cells, and this has given rise to the development and therapeutic application of a number of biological therapies, small molecules (such as Janus kinase inhibitors), and other immunomodulators. [2,3] Although biological therapies, such as anti-tumor necrosis factor (TNF-)α therapies, have been successfully introduced, up to 40% of patients with IBD are inadequate responders to primary anti-TNF-α treatment, and another 30% to 40% lose this response within 1 year of treatment [4,5].

As such, there is a significant, unmet need for new targeted therapies for TNF-inadequate responders [6,7]. Recent studies have demonstrated that stromal cells, in addition to intestinal epithelial and inflammatory cells, play an important role in the pathogenesis of IBD. Recent studies have shown that fibroblasts, which are the most abundant stromal cells, not only produce excessive ECM and cause fibrosis complications but also play an important role in maintaining tissue homeostasis by interacting with both epithelial and immune cells [8].

Over the past decade, rapid advances in RNA sequencing (RNA-seq), single-cell tissue profiling, and spatial transcriptomic techniques have enabled the unprecedented examination of stromal cells, including fibroblasts, in pathological tissues [9]. Here, we report a bulk transcriptomic study of inflamed IBD tissues using RNA sequencing (RNA-seq) to investigate the characteristics of inflammatory tissues and their associations with the response to biologic therapy.

## 2. Results

### 2.1. PCA of CD Inflamed Samples

Our 205 samples included 59 inflamed and 146 non-inflamed samples collected from 170 patients with CD. The PCA of our samples showed two distinct clusters of inflamed samples: one close to (Inflamed_1) and the other far from (Inflamed_2) the non-inflamed samples (Figure 1A). Similar trends were observed with both the dataset_1 and dataset_2 protocols. Between the two inflamed clusters, there were no statistically significant differences in the clinical characteristics between the two inflamed clusters, except for age at diagnosis (*p* < 0.0001), history of using immunosuppressants (*p* < 0.0001), and anti-TNF (*p* = 0.026) (Table 1).

Among 170 patients, 22 patients underwent anti-TNF treatment before sample collection (Table 1) and had response data. The samples collected from the remaining 148 individuals are pretreatment samples. Next, we investigated whether these two clusters were associated with the response to anti-TNF treatment in 22 inflamed samples from 22 patients who underwent anti-TNF treatment before sample collection. As previously described [10], a durable responder (DR) was defined as an individual maintaining their response to anti-TNF therapy for at least 24 months after initiation. A non-durable responder (NDR) was defined as someone with a non-response within 24 months of starting therapy, accompanied by an alteration in therapy (the addition or escalation of corticosteroids, switching to a different agent, or surgery).

Two inflamed clusters showed distinct patterns: the cluster near the non-inflamed samples (Inflamed_1) was enriched in DRs (5:1 and 6:1 in the dataset_1 and dataset_2, respectively), and the distant cluster (Inflamed_2) was relatively enriched in NDRs (4:3 and 1:1 in the dataset_1and dataset_2, respectively) (Figure 1B). The proportions of DRs and NDRs did not reach statistical significance between the two inflamed samples (*p* = 0.178).

### 2.2. Cell Fraction Using Cibersortx

The signature matrix generated from the Ileum scRNA-seq dataset contained 23 cell types and the expression profiles of 3543 genes. This ileum signature matrix was used in silico to deconvolve ileum cell subsets from our bulk RNA-seq using Cibersortx. Figure 2A shows the proportions of five major cell types in the Inflamed_1 (n = 26), Inflamed_2 (n = 12), and non-inflamed (n = 101) categories from dataset_1, as well as the Inflamed_1 (n = 19) and non-inflamed (n = 45) categories from dataset_2. Among the 23 cell types, activated fibroblasts, a type of stromal cells, were predicted to account for 1.37% in dataset_1Inflamed_1, 1.57% in dataset_2Inflamed_1, and 9.41% in dataset_1Inflamed_2. They were observed in an approximately six-fold higher proportion in Inflamed_2 compared to Inflamed_1 (*p* = 0.007, Figure 2B).

### 2.3. Validation

In the above results, it was confirmed that activated fibroblasts existed in a higher proportion in the Inflamed_2 tissue. To verify whether the same result held true in other datasets, two cohorts were used. First, the RISK (GSE134881) cohort was utilized. This cohort consisted of children who developed the disease after the age of 2; their pathophysiology is very similar to that observed in adults. Responders (DR) were defined as patients who achieved sustained steroid-free clinical remission between 18 and 24 months after disease diagnosis and received anti-TNF antibody therapy. In the PCA based on the log2 CPM expression matrix, it was not possible to distinguish between DRs and NDRs (Figure 3A). Using the ileum signature matrix as a reference, the cell fractions were predicted using Cibersortx for 25 DR samples and 34 NDR samples, resulting in proportions of activated fibroblasts of 0.85% in DRs and 1.49% in NDRs among the 23 cell types. Although the incidence of NDRs was approximately 1.5 times higher than that of DRs, it did not show statistical significance (*p* = 0.259) (Figure 3B).

Secondly, the GSE16879 cohort, which included patients with colon CD (CDc) and ileum CD (CDi) before and after anti-TNF therapy, as well as normal samples, was used [11]. The “before” samples underwent endoscopy and biopsy one week before the intravenous injection of anti-TNF. The “after” samples underwent a second endoscopy and biopsy 4 weeks after single anti-TNF therapy, 4 weeks after anti-TNF infusion, and at week 0, week 2, and week 6 after receiving anti-TNF therapy. In the PCA microarray expression values based on log2, the “before” and “after” samples of CDc and CDi were clearly distinguishable, but it was difficult to differentiate between “after” and “before” in terms of DRs and NDRs (Figure 3C). Cibersortx can be used regardless of the biological source, so it did not differentiate between colon and ileum tissues [12]. Using the ileum signature matrix as a reference, cell fractions were predicted using Cibersortx for 19 DR samples (11 colon, 8 ileum) and 17 NDR samples (7 colon, 10 ileum). Among all the cell types, the predicted proportion of activated fibroblasts in the CD inflamed “before” samples was 1.1% in 19 DR samples and 2.1% in 17 NDR samples, showing a 1.95-fold higher proportion in NDRs compared to DRs (*p* = 0.01). In the case of the “after” anti-TNF therapy category, the proportion was 0.2% in DR samples and 1.8% in NDR samples, indicating an approximately 7.49-fold higher proportion in NDRs (*p* = 0.01). In the CD normal samples, after anti-TNF therapy, the proportion of activated fibroblasts was 0.2%, similar to that in the DR samples (Figure 3D).

### 2.4. Differentially Expressed Genes in Inflammatory Tissues

A total of 3066 differentially expressed genes (DEGs) were identified with a log2 fold change (logFC) > 1 and a *p*-value < 0.05, comparing between Inflamed_1 and Inflamed_2 from dataset_1. Among these, 1373 genes were observed in Inflamed_2 (DEGs were not analyzed in Inflamed_2 samples from dataset_2 owing to the small sample size) (Appendix A). Since Inflamed_1 was closer to non-inflamed CD in the PCA, it is possible that the DEGs between inflamed and non-inflamed are included among the 3066 DEGs. To eliminate this possibility, the DEGs in inflamed and non-inflamed tissues were analyzed, resulting in the identification of 1594 inflamed DEGs and 246 unique DEGs in Inflamed_2 that did not overlap. Among the 246 DEGs, THBS2, CXCL5, EGFL6, and fibroblast activation protein (FAP) showed greater changes in the NDR samples in both the RISK and GSE16879 cohorts (Table 2).

Next, we used the David web service to identify pathway enrichment, focusing on the “FDR < 0.05” condition during our analysis of “Inflamed_1 vs. Inflamed_2” and “Inflamed vs. Non-inflamed.” Consequently, we found 70 KEGG pathways within the “Inflamed” category of “Inflamed vs. Non-inflamed” and 53 KEGG pathways within the “Inflamed_1” category of “Inflamed_1 vs. Inflamed_2.” Moreover, it was confirmed that the 53 pathways identified in the “Inflamed_2” category were encompassed within the 70 pathways identified in the “Inflamed” category (Appendix A).

## 3. Discussion

Here, we report a bulk transcriptomic study of the inflamed and non-inflamed tissues of patients with IBD using RNA-seq. The PCA plots of our bulk RNA-seq data displayed two distinct clusters of CD-inflamed samples: one enriched with NDRs (Inflamed_2) and the other enriched with DRs (Inflamed_1) in anti-TNF therapy. The Cibersortx results showed that the average cell fraction of activated fibroblasts, a type of stromal cell, was approximately six times higher in Inflamed_2 than in Inflamed_1. In our validation with two other CD public gene expression datasets, the NDR samples in the RISK cohort had approximately 1.5 times higher levels of activated fibroblasts than the DR samples, although it did not show statistical significance. In another cohort (GSE16879 dataset), activated fibroblasts were enriched in NDRs over DRs by 1.9 times before treatment, and this ratio increased to 7.5 times after treatment. In the DEG analysis, FAP was included in 246 unique DEGs in Inflamed_2 in our dataset and was also overexpressed in the NDR in both the RISK and GSE16879 datasets.

Fibroblasts are mesenchymal cells that compose the stroma of organ tissues [9]. Several genes and protein markers, such as COL1A2 and PDGFRA, have been used to identify cells of non-hematopoietic, non-epithelial, and non-endothelial lineages; there are few or no fibroblast-specific markers. Moreover, the function of many fibroblast markers is not completely understood. However, the advent of transcriptomic profiling through RNA sequencing and single-cell RNA-seq (scRNA-seq) has enabled the in-depth examination of fibroblasts in pathologic tissue and indicates that fibroblasts include diverse cell types based on their developmental origin, anatomic location, and function [9].

In one of the first IBD studies to examine human colonic fibroblasts, researchers performed scRNA-seq on colonic mesenchymal cells from patients with UC [13]. They identified four colonic stromal fibroblast populations (referred to as S1–S4) and noted the emergence of an activated fibroblast subset (S4) that was highly expanded in UC and was characterized by the expression of IL-6, MHC class II invariant chain (CD74), IL-33, and homeostatic cytokines mediating lymphocyte recruitment and retention (CCL19 and CCL21). Other studies on IBD [6,7,14] have described the association between fibroblasts and the response to anti-TNF therapy. West et al. [14] described that inflammation-associated fibroblasts were drastically expanded in the inflamed tissues of some patients with CD and UC. Using bulk expression data to define TNF resistance, the authors found a strong enrichment of the TNF resistance signature in inflammation-associated fibroblasts. Smilllie et al. [7] examined human colonic fibroblasts in patients with UC using scRNA-seq and identified multiple fibroblast subsets characterized by the differential expression of Wnt and BMP signaling genes. They found that although most fibroblast subsets were present in both healthy individuals and patients with UC, a subset termed inflammation-associated fibroblasts (IAFs) were expanded 189-fold in the inflamed tissues of some patients. They scored the cell subsets for gene signatures of anti-TNF resistance and sensitivity based on a meta-analysis of bulk expression data from 60 responders and 57 non-responders to therapy [15]. The drug resistance signature was strongly enriched in the IAFs. In a separate scRNA-seq study, Martin et al. [6] identified two fibroblast subtypes in patients with CD. Using a unique signature that incorporated the activated fibroblast gene expression profile, the authors found an association between the enrichment of this signature in patients before treatment and their resistance to anti-TNF therapy.

Including our results, as studies of fibroblasts in IBDs point to the role of inflammatory (or activated) fibroblasts in TNF-inadequate responders, it is tempting to speculate that therapies targeting inflammatory (or activated) fibroblasts in such patients could help to overcome inadequate responses. In a mouse model, the selective depletion of stromal cells expressing fibroblast activation proteins (FAP) using diphtheria toxin abrogated inflammatory arthritis [16]. Recently, Wang et al. [17] integrated eight tissue transcriptomic datasets from patients with IBD treated with anti-TNF-α therapies along with single-cell RNA-seq data from UC to identify TNF-inadequate response mechanisms. They included the RISK and GSE16879 datasets we used for the validation of our results. RNA-seq data from upadacitinib (Janus kinase (JAK) 1 inhibitor) and risankizumab (IL-23 inhibitor) phase 2 CD clinical trials were used to support the clinical response to upadacitinib and risankizumab in IBD TNF-inadequate responders. Inflammatory fibroblasts were the only fibroblast cell type that was significantly reduced in JAK1-response patients following upadacitinib treatment. Thus, they suggested that upadacitinib may block pathways that remain active in patients with IBD who do not respond to anti-TNF therapy. 

Among 246 unique DEGs in the Inflamed_2 samples, THBS2, CXCL5, EGFL6, and FAP also showed greater changes in the NDR samples in both the RISK and GSE16879 cohorts. FAP, a protein expressed in response to triggers like epithelial cancers, wounds, and chronic inflammation, plays a pivotal role in activating fibroblasts [18]. In mouse models, selectively depleting FAP has been shown to prevent the development of inflammatory arthritis [16]. To understand the significance of THBS2, in a previous study, researchers conducted functional gel contraction experiments on fibroblasts to assess their ability to remodel the extracellular matrix. The results demonstrated that purified THBS2 significantly boosted the gel contraction capacity of fibroblasts. Additionally, reducing THBS2 levels was found to diminish the ability of cancer cells to activate pulmonary fibroblasts [19]. In scRNA-seq, CXCL5 serves as a marker gene representing activated fibroblasts within the GIMAT [6]. However, specific information about the relationship between fibroblasts and EGFL6 remains limited.

The strength of our research lies in the fact that while most studies have elucidated the role of fibroblasts by comparing inflammatory and non-inflammatory tissues in IBD, we confirmed that they can be divided into two groups within the inflammatory tissue. We have shown that in some patients with CD with a high number of activated fibroblasts in their inflammatory tissues, the TNF response can be reduced. In other words, we demonstrated the potential for predicting the response to anti-TNF therapy based on the characteristics of inflammatory tissues. Furthermore, we confirmed that our results are similar to those of other studies that used single-cell sequencing by utilizing bulk data through the Cibersort method with single-cell data as a reference. We found several genes, such as THBS2, CXCL5, EGFL6, and FAP, that are associated with NDRs to anti-TNF therapy. As FAP is an important marker of activated fibroblasts, it may be possible to predict the response to anti-TNF therapy in advance if these markers are checked in the inflammatory tissue before anti-TNF therapy. Additional research is required to comprehend the underlying mechanism behind insufficient reactions to anti-TNF therapy and their connections with these genes.

A limitation of our study is that, as clinical data regarding response to anti-TNF therapy were only available for a few patients, the proportion of DRs and NDRs did not reach a statistically significant difference between the two inflamed samples. However, we validated our results using other cohorts. In addition, it will be possible to prospectively predict and validate the response before using anti-TNF or other biologics such as a JAK 1 inhibitor or IL-23 inhibitors in the remaining patients in our cohort. Moreover, the functions of many fibroblast markers are not completely understood; thus, different terms are used to describe specific populations of fibroblasts such as inflammatory fibroblasts [7,14] and activated fibroblasts [7,13] in IBD studies. The IAFs in a previous UC study expressed FAP, a marker of cancer-associated fibroblasts, and vimentin (VIM) as a marker validated experimentally in IAFs. Within the IAF subgroup, there were inflammatory fibroblasts, and podoplanin (PDPN) was exclusively used as a marker for inflammatory fibroblasts among several other fibroblasts. The activated fibroblasts referenced in our Cibersort method from a previous CD study [6] served as representative markers, including THY1 and PDPN. Referring to a study by Tuley et al. [20], FAP, PDPN, and VIM were used as markers of gene expression. Consequently, the inflamed fibroblasts from the UC study [7], which are a sub-cell type of stromal cells, and the activated fibroblasts from the CD study [6] appear to be the same because they share PDPN and VIM as markers. We believe that using the same terminology in future studies will help to reduce confusion.

In conclusion, activated fibroblasts may play a role in reducing resistance to anti-TNF therapy. Characterizing fibroblasts in heterogeneous inflamed tissues at diagnosis could, therefore, be helpful in selecting patients who are likely to respond to anti-TNF therapy.

## 4. Materials and Methods

### 4.1. Study Samples

Patients were included from the IMPACT (identification of the mechanism of the occurrence and progression of CD through integrated analysis on both genetic and environmental factors) study cohort, which is a prospective multicenter study that was established in Korea in 2017. A total of 16 university hospitals participated in this study, and clinical data and biological specimens (including blood, stool, and tissue specimens) of patients with CD, who were newly diagnosed or followed up within the institutions, were collected as previously described [21,22,23]. Ethical approval for the present study was provided by the Institutional Review Boards of Kangbuk Samsung Hospital (KBSMC 2016-07-029 and KBSMC 2020-05-021) and each participating center. Written consent was obtained from all participants after the nature and possible consequences of the study were explained to them.

### 4.2. Sample Collection

RNA-seq data were collected from the ileocolonoscopic tissue samples of patients with CD, as described previously [23]. Patients diagnosed with CD undergo an ileocolonoscopy as part of their routine medical care. A biopsy of the most severe ileal or colonic inflammatory lesions and normal lesions was performed. The lesions were assessed as inflamed or normal based on the endoscopic findings at the time of collection. Only normal tissue was obtained when patients’ colonoscopic findings showed endoscopic remission, and it was mostly obtained from the ileocecal valve.

### 4.3. Sample Preparation, Library Construction, and RNA-Sequencing

The RNA extraction process is described in our previous paper [23]. Approximately 1 μg of total RNA was used for library construction with either the MGIEasy RNA Directional Library Prep Set (MGI Tech, Shenzhen, China) or TruSeq Stranded Total RNA Library Prep Kit (Illumina, San Diego, CA, USA). Next, paired-end sequencing was performed using either MGI DNBSEQ-g400 (“dataset_1”) or the Illumina NovaSeq 6000 System (“dataset_2”) according to the respective manufacturer’s instructions, depending on the library preparation process. The RNA extraction process was conducted at the Functional Genome Institute, PDXen Biosystems, Inc. (Daejeon, Republic of Korea), while the library preparation and sequencing processes were conducted at the Personalized Medicine Research Center, Korea Research Institute of Bioscience and Biotechnology (KRIBB).

### 4.4. PCA and Cibersortx

RNA-seq read quality control and mapping processes were conducted separately for samples prepared using different protocols and platforms. The reads were mapped to the GRCh38 reference genome using HISAT2 [24]. Read assignment was performed using feature counts [25] based on the Ensembl gene model (Release 100). The quantified raw counts were normalized to counts per million (cpm), based on the trimmed mean of M-values method using EdgeR [26] in the Bioconductor framework. Principal component analysis was performed with the log-transformed CPM values using EdgeR.

Biopsy samples were collected via colonoscopy from the inflammatory and non-inflammatory lesions of patients with CD-comprised various cell types. Neither tissue dissection nor cell purification was performed. Instead, in silico cytometry was performed to estimate the cellular heterogeneity using Cibersortx [12], an in silico cell fraction estimation algorithm. Figure 4 shows an overview of the pipeline. Cibersortx uses a machine learning algorithm to infer the composition of each cell type in bulk samples based on a signature matrix derived from a reference expression dataset that has been measured in relevant tissues using single-cell RNA-seq (scRNA-seq) technology.

To construct reference profiles, we downloaded the scRNA-seq (GSE134809) dataset with measurements from the ileum of patients with CD prior to anti-TNF treatment. Two samples with low cell numbers or similar profiles between the inflamed and non-inflamed tissues were rejected, resulting in nine inflamed and non-inflamed pairs included in the analysis. The scRNA-seq dataset was analyzed using the Seurat v4 [27] R package. The preprocessing workflow (2000 > nCount_RNA > 1000%, percentage mitochondrial mRNA < 25%, epithelial cells < 1%, and red blood cells < 10) yielded 66,978 cells. Subsequently, the FindVariableFeatures, FindIntegrationAnchors, and IntegrateData functions in Seurat were applied in succession, resulting in 31,566 cells, which were annotated with ScType [28]. The resulting gene expression matrix was used in Cibersortx to construct a signature matrix and correct batch effects using the S-mode.

### 4.5. Statistical Analysis

A descriptive analysis was performed using means and standard deviations for continuous measures and percentages for categorical measures. The chi-square test or Fisher’s exact test was used for the analysis of categorical variables, and continuous variables were analyzed using Student’s *t*-test. Logistic regression was used to assess the statistical significance of the impact of activated fibroblasts on the inflamed samples.

## Figures and Tables

**Figure 1 ijms-24-14799-f001:**
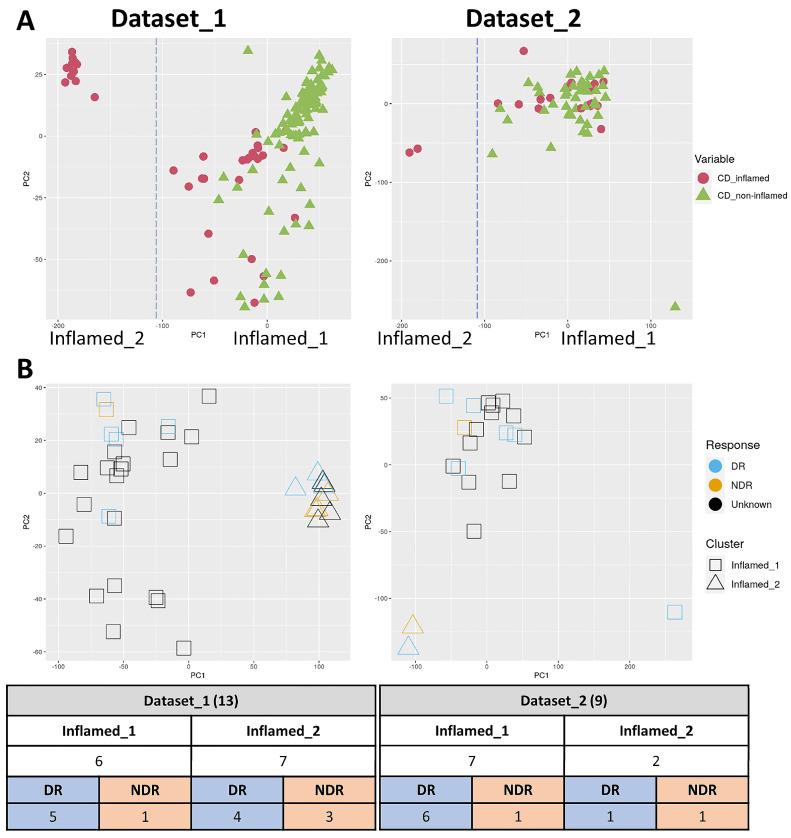
PCA plots of the two datasets, dataset_1 and dataset_2. (**A**) The PCA visualization was used to examine the relationship between the inflamed samples and non-inflamed samples, resulting in the categorization of the CD_inflamed samples into two groups: “Inflamed_1” and “Inflamed_2”. (**B**) The response to anti-TNF therapy depicted for the inflamed samples. Cases with a durable response (DR) are shown in blue, while cases involving a non-durable responder (NDR) are shown in orange. Among the “Inflamed_1” samples in dataset_1, there were 5 cases of DR and 1 case of a NDR. For the “Inflamed_2” samples in dataset_1, there were 4 DR cases and 3 NDR cases. In dataset_2, among the “Inflamed_1” samples, there were 6 DR cases and 1 NDR case. Additionally, among the “Inflamed_2” samples in dataset_2, there was 1 DR case and 1 NDR case.

**Figure 2 ijms-24-14799-f002:**
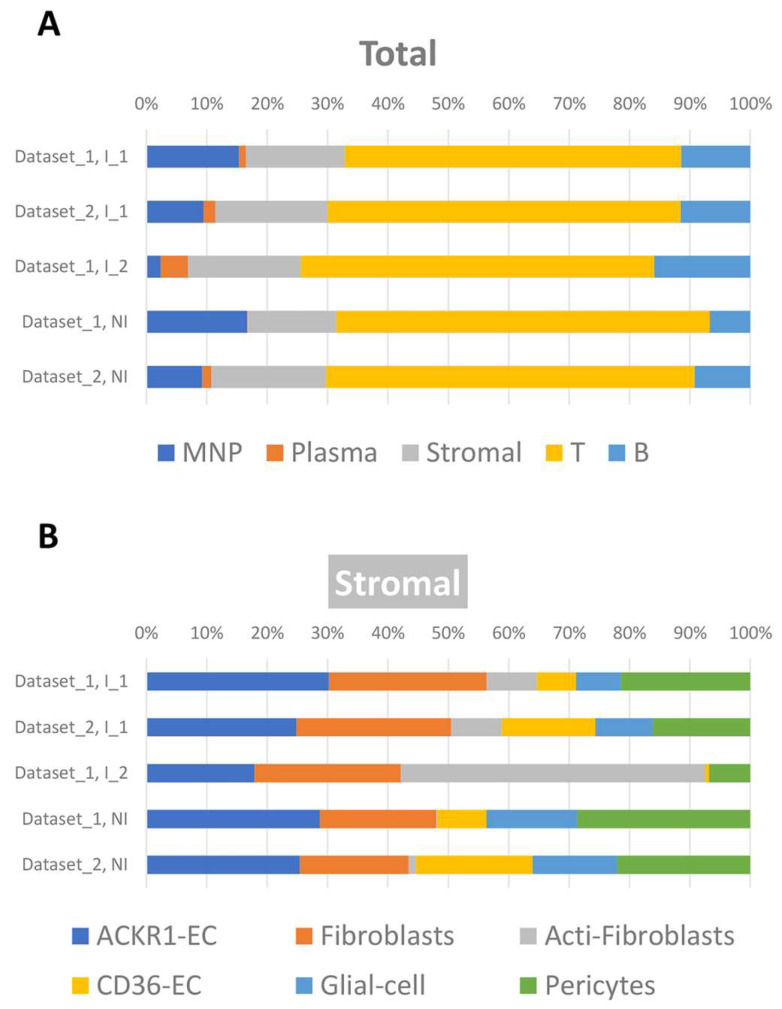
Cibersortx visualization of the relative proportions of different cell types. (**A**) Cell types are categorized into 5 main types: mononuclear phagocytes (MNP) in blue, plasma cells in orange, stromal cells in gray, T cells in yellow, and B cells in blue. (**B**) Only the Stromal cell types are depicted: atypical chemokine receptor 1+ endothelial cells (ACKR1-EC) in blue, fibroblasts in orange, activated fibroblasts (Acti-Fibroblasts) in gray, CD36+ endothelial cells (CD36-EC) in yellow, glial cells in sky blue, and pericytes in green. In the stacked bar plots, the top two bars depict the mean proportions of the “Inflamed_1 (I_1)” samples in both dataset_1 and dataset_2. The mean proportion of the “Inflamed_2 (I_2)” samples in dataset_1 is plotted in the middle row, while the corresponding plot of dataset_2 is omitted, as only two samples belong to “Inflamed_2”. The bottom two bars represent the non-inflamed (NI) samples.

**Figure 3 ijms-24-14799-f003:**
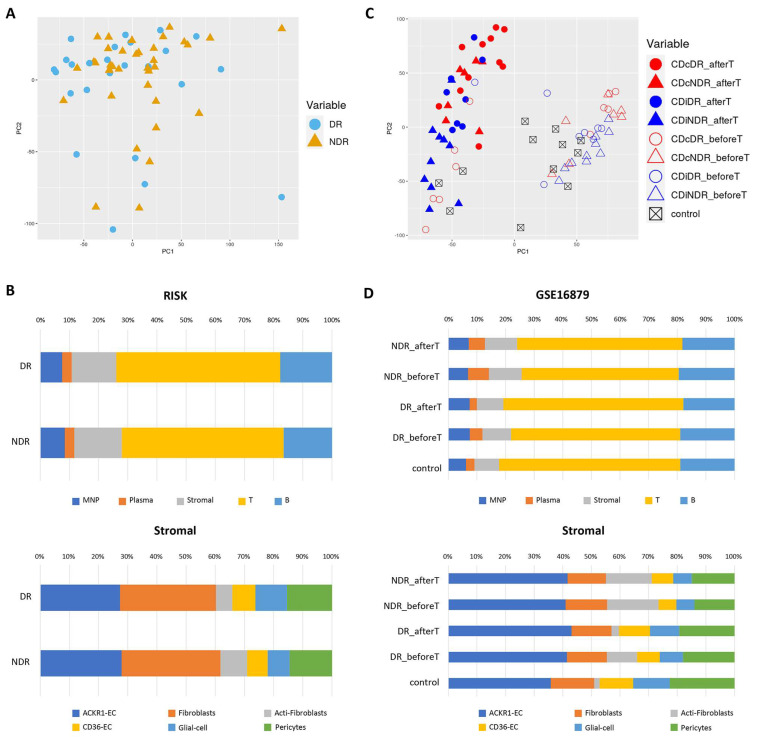
PCA and Cibersortx visualizations of the validation datasets. For validation purposes, we utilized two datasets. The RISK cohort consisted of CD inflamed samples, with 34 non-durable responders (NDRs) and 25 durable responders (DRs). On the other hand, the GSE16879 cohort included CD inflamed samples before and after anti-TNF therapy, with 17 non-durable responders (NDRs), 19 durable responders (DRs), and 12 control samples. (**A**) The PCA plot of the RISK cohort is shown: the NDR samples are depicted with orange triangles, the DR samples with blue circles. (**B**) The stacked bar plot illustrates the estimated proportions of different cell types in the RISK cohort, obtained through Cibersortx analysis. (**C**) The PCA plot of GSE16879 is shown, where the colon CD (CDc) and ileum CD (CDi) samples are indicated in red and blue, respectively. The filled and empty symbols represent the conditions after and before anti-TNF therapy, respectively. Additionally, the control samples are marked with black boxes. (**D**) The stacked bar plot presents the mean proportions of different cell types in the GSE16879 cohort, obtained through Cibersortx analysis.

**Figure 4 ijms-24-14799-f004:**
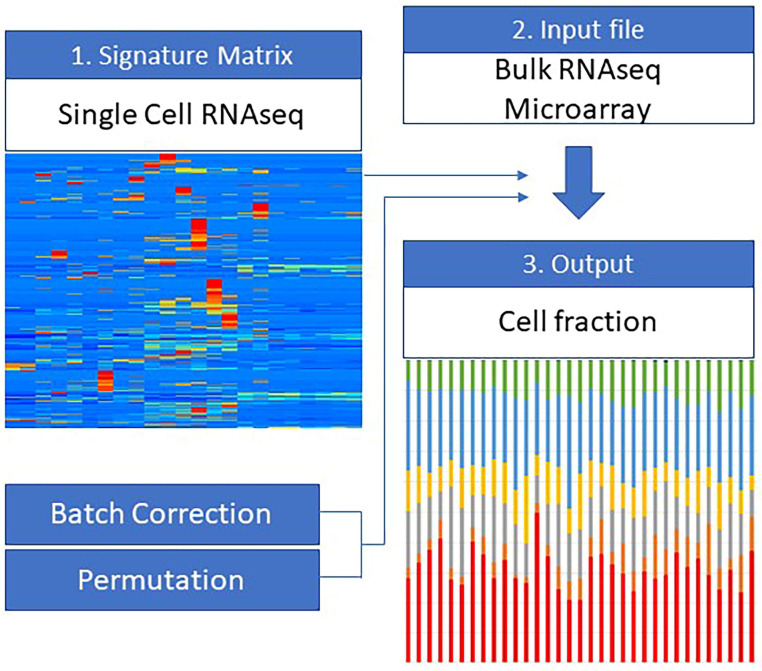
Illustration of the cellular heterogeneity analysis pipeline using Cibersortx. Cibersortx utilizes machine learning algorithms to estimate the relative proportions of various cell types from bulk RNA sequencing data. It identifies cell types using a signature matrix and corrects the significance of the results through batch correction and permutation techniques, ensuring accurate and reliable outcomes.

**Table 1 ijms-24-14799-t001:** Clinical characteristic of patients with Crohn’s disease.

	Inflamed_1 (n = 45)	Inflamed_2 (n = 14)	*p*-Value
Age at diagnosis, year (SD)	28.1 (11.9)	20.5 (4.4)	<0.001
Gender, male (%)	35 (71.4%)	10 (71.4%)	0.660
History of smoking, n (%)	5 (11.1%)	3 (21.4%)	0.650
Family history of IBD, n (%)	5 (11.1%)	1 (7.1%)	0.650
Disease duration, year (SD)	3 (3.9)	4.4 (4.1)	0.270
Disease location, n (%)			0.070
Ileal	9 (20%)	1 (7.1%)	
Colonic	12 (26.7%)	2 (14.3%)	
Ileocolonic	24 (53.3%)	11 (78.6%)	
Upper GI involvement, n (%)	0 (0%)	3 (21.4%)	0.080
Disease behavior, n (%)			0.890
Inflammatory	34 (75.6%)	11 (78.6%)	
Stricturing	8 (17.8%)	2 (14.3%)	
Penetrating	3 (6.7%)	1 (7.1%)	
Perianal disease, n (%)	13 (28.9%)	5 (35.7%)	0.650
Previous treatment			
Immunosuppressants, n (%)	35 (77.8%)	14 (100%)	<0.001
Anti-TNF	13 (28.9%)	9 (64.3%)	0.026
Intestinal resection, n (%)	11 (24.4%)	7 (50%)	0.110

**Table 2 ijms-24-14799-t002:** Key differentially expressed genes in various cohorts.

DEGs	This Study	RISK		GSE16879			
	Inflamed_2 vs. Inflamed_1	Before Anti-TNF NDR vs. DR		After Anti-TNF NDR vs. DR		Before Anti-TNF NDR vs. DR	
	logFC, FDR < 0.05	logFC	*p*-Value	logFC	*p*-Value	logFC	*p*-Value
THBS2	3.40	2.61	<0.001	1.18	<0.001	0.71	<0.001
CXCL5	3.27	1.24	0.016	1.19	<0.001	0.57	<0.001
EGFL6	3.13	1.41	0.013	1.04	<0.001	0.58	<0.001
FAP	2.92	1.49	0.007	1.74	<0.001	0.75	<0.001

DEGs, Differentially Expressed Genes; NDR, N on-durable responders; DR, Durable responders; logFC, log2 fold change; THBS2, Thrombospondin 2; CXCL5, CXC motif chemokine ligand 5; EGFL6, EGF like domain multiple 6; FAP, Fibroblast activation protein.

## Data Availability

All primary RNA sequencing data were deposited in the Korean Nucleotide Archive (KoNA) database under the BioProject accession ID KAP220153 (https://www.kobic.re.kr/kona/search_bioproject?bioproject_id=KAP220153, accessed on 31 May 2022).

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
