# Peer review of "Enrichment of Activated Fibroblasts as a Potential Biomarker for a Non-Durable Response to Anti-Tumor Necrosis Factor Therapy in Patients with Crohn’s Disease"

_ijms, 2023, doi:10.3390/ijms241914799_

Round 1

Reviewer 1 Report

Soo-Kyung et all are peorting about a mechanism to understand responsiveness against anti-TNf therapy. They used a cohort of over 200 patients (or samples) from bulk-RNAseq data was obtained. Biopsies were of ileocolonic origin. It remains unclear how many samples were actually used, when (in what part of treatment or pretreatment period) they were taken etc.  I addition, why no similar number of inflamed vs non-inflamed? What about other characteristics - previous therapies etc? Why bulk and non sc-RNAseq? In the results, no description of statistical power (which is probably low).

To be very honest, the methods and results are at this point not acceptable. 

Multiple repetitions in intro and abstract, the language is overall good but too complicated at times.

Reviewer 2 Report

The study of Park et al. “Enrichment of activated fibroblast as a potential biomarker for non-durable response to anti-tumor necrosis factor therapy in patients with Crohn’s disease” focuses on the investigation of the response to anti-tumor necrosis factor (anti-TNF) treatment depending on the inflammatory tissue characteristics in Crohn’s disease using Bulk RNA sequencing approach with CiberSortx. Authors demonstrated that cell fraction of activated fibroblasts was higher in inflamed colon tissue of patients non-durable responders than in inflamed colon tissue of anti-TNF durable responders. Fibroblast activation protein was overexpressed with similar correlations.

Major points:

1. The authors propose to characterize colon tissue fibroblasts by their heterogeneity at diagnosis to predict the response to therapy of patients with Crohn’s disease. By what method can this characterization be carried out in routine clinical practice? What specific characteristics of fibroblasts do you intend to use?

2. The authors collected inflamed and non-inflamed material from the same patient. Is it reliable to use endoscopic findings alone to differentiate between inflamed and non-inflamed tissue? Do you consider it necessary to monitor endoscopy data by histological examination of the tissue?

3. The authors did not perform functional annotation of DEGs and did not characterize the key genes (THBS2, CXCL5, EGFL6, and FAP), whose expression changes most.

Minor points: typos on lines 90, 140, 152, 160, 211, 229, 247.

Round 2

Reviewer 2 Report

The authors successfully provided some clarifications in the manuscript that made it more clear and scientific sound.